# T_3_ Intratracheal Therapy Alleviates Pulmonary Pathology in an Elastase-Induced Emphysema-Dominant COPD Mouse Model

**DOI:** 10.3390/antiox13010030

**Published:** 2023-12-22

**Authors:** Noriki Takahashi, Ryunosuke Nakashima, Aoi Nasu, Megumi Hayashi, Haruka Fujikawa, Taisei Kawakami, Yuka Eto, Tomoki Kishimoto, Ayami Fukuyama, Choyo Ogasawara, Keisuke Kawano, Yukio Fujiwara, Mary Ann Suico, Hirofumi Kai, Tsuyoshi Shuto

**Affiliations:** 1Department of Molecular Medicine, Graduate School of Pharmaceutical Sciences, Kumamoto University, 5-1 Oe-honmachi, Chuo-ku, Kumamoto 862-0973, Japan; noriki.t07@outlook.jp (N.T.); a.nasu@po.nippon-shinyaku.co.jp (A.N.); 236y2014@st.kumamoto-u.ac.jp (M.H.); haruka-fujikawa@kke.co.jp (H.F.); 181p1015@st.kumamoto-u.ac.jp (T.K.); kawakami1505@gmail.com (T.K.); 191p1050@st.kumamoto-u.ac.jp (A.F.); 236y1008@st.kumamoto-u.ac.jp (C.O.); 207p2006@st.kumamoto-u.ac.jp (K.K.); mann@gpo.kumamoto-u.ac.jp (M.A.S.); hirokai@gpo.kumamoto-u.ac.jp (H.K.); 2Program for Leading Graduate Schools “HIGO (Health Life Science: Interdisciplinary and Global Oriented) Program”, Graduate School of Pharmaceutical Sciences, Kumamoto University, 5-1 Oe-honmachi, Chuo-ku, Kumamoto 862-0973, Japan; 3Department of Cell Pathology, Graduate School of Medical Sciences, Kumamoto University, 1-1-1, Honjo, Kumamoto Chuo-ku, Kumamoto 860-8556, Japan; fuji-y@kumamoto-u.ac.jp; 4Global Center for Natural Resources Sciences, Faculty of Life Sciences, Kumamoto University, 5-1 Oe-Honmachi, Chuo-ku, Kumamoto 862-0973, Japan

**Keywords:** thyroid hormone (T_3_), chronic obstructive pulmonary disease (COPD), disease type classification, mitochondrial function, antioxidant effect, pulmo-modulatory factors

## Abstract

Chronic obstructive pulmonary disease (COPD) is a complex pulmonary condition characterized by bronchitis, emphysema, and mucus stasis. Due to the variability in symptoms among patients, traditional approaches to treating COPD as a singular disease are limited. This led us to focus on phenotype/endotype classifications. In this study, we explore the potential therapeutic role of thyroid hormone (T_3_) by using mouse models: emphysema-dominant elastase-induced COPD and airway-dominant C57BL/6-βENaC-Tg to represent different types of the disease. Here, we showed that intratracheal T_3_ treatment (40, 80 μg/kg, *i.t.*, every other day) resulted in significant improvements regarding emphysema and the enhancement of respiratory function in the elastase-induced COPD model. T_3_-dependent improvement is likely linked to the up-regulation of *Ppargc1a*, a master regulator of mitochondrial biogenesis, and *Gclm*, a factor associated with oxidative stress. Conversely, neither short- nor long-term T_3_ treatments improved COPD pathology in the C57BL/6-βENaC-Tg mice. Because the up-regulation of extrathyroidal T_3_-producing enzyme *Dio2*, which is also considered a marker of T_3_ requirement, was specifically observed in elastase-induced COPD lungs, these results demonstrate that exogenous T_3_ supplementation may have therapeutic potential for acute but not chronic COPD exacerbation. Moreover, this study highlights the relevance of considering not only COPD phenotypes but also COPD endotypes (expression levels of *Ppargc1a* and/or *Dio2*) in the research and development of better treatment approaches for COPD.

## 1. Introduction

Chronic obstructive pulmonary disease (COPD) is a refractory pulmonary disease and one of the major world health problems [1]. Its main symptoms are chronic airway inflammation, emphysema, mucus stasis in the airway, and a decline in respiratory function [2,3]. COPD develops via the long-term inhalation of oxidants such as tobacco smoke and harmful air pollutants [4]. In addition, its onset intricately relates to various risk factors such as senescence, protease/antiprotease balance, environment, and genetic factors, among others [2]. Therefore, symptom variability in COPD patients can be observed [5]. However, the current treatment strategy focuses on common symptomatic therapy, and there is no personalized therapy for COPD. In order to establish novel therapies for COPD, it is important to focus not only on the phenotype but also on the molecular mechanisms underlying the clinical features, called endotypes [6]. In fact, phenotype/endotype classification has been considered for a better understanding of the disease mechanisms in asthma [7], atopic dermatitis [8], and sepsis [9].

COPD is classified approximately into two disease phenotypes: emphysema- or airway-dominant COPD [10]. Emphysema-dominant COPD patients present severe emphysema and are usually thin. On the other hand, airway-dominant COPD patients present airway remodeling and mucus stasis. Despite this knowledge, there are only a few basic studies based on the disease type classification of COPD. In this study, we evaluated a therapeutic candidate compound of COPD in both emphysema- and airway-dominant COPD models, represented by the elastase-induced COPD model [11] and the C57BL/6J-βENaC-Tg mouse [12], respectively. In the elastase-induced COPD model, transient and acute inflammation is induced 1 day after elastase is administered, and severe emphysema develops 3 weeks later. This model has mostly been used in many studies on COPD. On the other hand, we and others established a new COPD mouse model, C57BL/6J-βENaC-Tg mice [12]. This mouse model overexpresses the β subunit of the epithelial sodium ion channel (ENaC), a major regulator of salt and water reabsorption, specifically in airway epithelial cells. An over-influx of sodium ions into the intracellular compartment decreases periciliary liquid and induces mucus stasis. In addition, this mouse model exhibits a number of the chronic clinical symptoms of COPD, such as airway inflammation, emphysema, and a decline in respiratory function. Notably, by using these two phenotypically/endotypically different models, we have identified unique modulatory factors that positively and negatively affect COPD pathogenesis in a phenotype/endotype-specific manner [12,13,14,15]. Here, we focused on another possible pulmo-modulatory factor 3,3′,5-Triiodo-L-Thyronine (T_3_), one of the thyroid hormones that has been reported to have various physiological activities, such as regulating systemic metabolism [16], antioxidant effects [17], and anti-inflammation effects.

Thyroid hormones are synthesized in the thyroid gland and are normally secreted as T_4_ [18]. T_4_ is converted into the active T_3_ by DIO2 (Type II iodothyronine deiodinase) in peripheral tissues, and T_3_ works as the main hormone. In the lungs, thyroid hormone has long been known to act as an essential factor for development and growth [19]. Recently, the increased expression of *Dio2* and decreased expression of *Ppargc1a* (peroxisome proliferator-activated receptor gamma coactivator 1-alpha), a master regulator of mitochondrial biogenesis, had been shown in a mouse model of refractory lung disease, idiopathic pulmonary fibrosis (IPF) [20]. Notably, the intratracheal administration of T_3_ ameliorated lung fibrosis by improving mitochondrial function. Moreover, another report showed the correlation between thyroid function and respiratory function in COPD patients [21]. Consistently, others also showed an increased level of DIO2 [22], as well as mitochondrial dysfunction [23], in COPD patients. However, the role of T_3_ is unknown in COPD pathology. Thus, we investigated the expression levels of *Dio2* and *Ppargc1a* and the effect of the intratracheal administration of T_3_ on COPD mouse models.

## 2. Materials and Methods

### 2.1. Animals

We used age-matched female C57BL/6J mice or C57BL/6J-βENaC-Tg mice in all animal experiments. All mice were bred in a 12 h light/dark cycle and fed with normal chow (Oriental Yeast, Tokyo, Japan) ad libitum in the animal facility at Kumamoto University. All experiments were performed according to the protocols approved by the Animal Welfare Committee of Kumamoto University (No. A2020-066), and all methods were performed in accordance with the guidelines and regulation of the animal facility center of Kumamoto University. C57BL/6J mice were purchased from Charles River Laboratories (Atsugi, Japan). The production of C57BL/6J-βENaC-Tg mice was previously reported [12], and gene-edited mice were stably provided by using an established reproductive technique at the Center for Animal Resources and Development (CARD) at Kumamoto University.

### 2.2. Elastase-Induced COPD Model Mice

We intratracheally administered elastase dissolved in saline (1 mg/mL) from porcine pancreas (E0127, Sigma-Aldrich Japan, Tokyo, Japan) (2 mg/kg) to 13-week-old female C57BL/6J mice. The day after the injection of elastase, inflammation was induced, and 3 weeks after injection, emphysema was induced.

### 2.3. Administration of 3,3′,5-Triiodo-L-Thyronine (T_3_)

3,3′,5-Triiodo-L-Thyronine (T_3_) was purchased from Sigma (T2877, Sigma-Aldrich, St. Loius, MO, USA) and dissolved in saline (50 or 100 μg/mL). We intratracheally administered T_3_ to C57BL/6J-βENaC-Tg mice (for 12 days or 2 months, every other day, 40 or 80 μg/kg) and to elastase-induced COPD mice (one injection or for 4, 7, 21 days, every other day, 40 or 80 μg/kg). We intratracheally administered saline as a vehicle to female C57BL/6J-βENaC-Tg mice or C57BL/6J mice. We used aerosol sprayers (KN-34700-1, Natsume Seisakusho, Tokyo, Japan) for the intratracheal administration for 2 months to minimize damage to the trachea.

### 2.4. Measurement of Pulmonary Mechanics and Function by flexiVent

To measure the pulmonary mechanics and function and assess the effect of administration of T_3_, we used flexiVent (SCIREQ, Montreal, QC, Canada), which is a computer-controlled small-animal ventilator, according to the manufacturer’s protocol. Briefly, mice were anesthetized with three types of mixed anesthesia (0.75 mg/kg of medetomidine (Nippon Zenyaku Kogyo Co., Ltd., Fukushima, Japan), 4 mg/kg of midazolam (Sandoz K. K., Tokyo, Japan), and 5 mg/kg of butorphanol (Meiji Animal Health Co., Ltd., Kumamoto, Japan)), and an 18-gauge needle was inserted into the trachea. Mice were ventilated using the flexiVent system with an average breathing frequency of 150 breaths/min. Inspiratory capacity (IC) was calculated by using a ventilation pattern: deep inflation. Pulmonary mechanics (compliance and elastance) were measured by using the forced oscillation technique, and pulmonary function (FEV0.1: forced expiratory volume in 0.1 s; FVC: forced vital capacity; FEV0.1%: forced expiratory volume % in 0.1 s) was determined by using a negative pressure reservoir connected to the flexiVent system and the flexiVent software, as described in a previous report [12].

### 2.5. Sample Collection

After the assessment of pulmonary mechanics and function, the mice were sacrificed for histological and biochemical analyses. Tissue sample of the left lobe of the lung was fixed in 10% formalin and then condensed in paraffin using histoprocessor Leica ASP300S (Bensheim, Germany). Tissue samples soaked in paraffin were wrapped in Leica EG1160 (Bensheim, Germany), and histological sections were prepared with a rotary microtome Leica RT2125RT (Bensheim, Germany).

### 2.6. Morphological Staining of Lung Tissue Section and Assessment of Emphysema

Histological sections were stained with periodic acid-Schiff (PAS) (Sigma-Aldrich) and alcian blue (pH 2.5) (Nacalai Tesque, Kyoto, Japan) to observe mucus-secreting goblet cells. These samples were counter-stained with hematoxylin (Sigma-Aldrich) and visualized using a BZ-X710 microscope (KEYENCE, Osaka, Japan) to assess lung morphology. These stained tissue sections were captured by BZ-X710 in 10 parts (3 upper, 4 middle, and 3 lower) per sample. The mean linear intercepts (MLI) were calculated as described in a previous report [11].

### 2.7. Image-Based Automatic Measurement of Alveolar Morphological Information

Images of whole lungs were captured using a BZ-X710 microscope, and alveolar morphological parameters, such as alveolar area, alveolar perimeter, (major axis + minor axis)/2, and ferret diameter, were automatically analyzed by using images with the BZ-Analyzer cell count software. Details of the auto-measure method were described in a previous report [11].

### 2.8. RNA Isolation, cDNA Synthesis, and Real-Time PCR

Total RNA from the mice lungs was isolated using RNAiso Plus (TaKaRa, Shiga, Japan), and cDNA was synthesized using PrimeScript RT reagent kit (TaKaRa). Real-time quantitative RT-PCR was performed using TB Green Premix Ex Taq II (TaKaRa) in CFX Connect systems (Bio-Rad, Richmond, BC, Canada). The relative quantity of target genes’ mRNA was normalized using mouse *Gapdh* or *18s* as the internal control, and this was expressed as the relative quantity of target genes’ mRNA (fold induction). The primer sets for real-time PCR are listed in Table 1.

### 2.9. Immunoblotting

The procedure for immunoblotting was previously described [14]. The primary antibodies used were mouse anti-PGC1α (sc-517380, Santa Cruz Biotechnology, Santa Cruz, CA, USA), rabbit anti-DIO2 (26513-1-AP, Proteintech, Rosemont, IL, USA), and rat anti-HSC70 (ADI-SPA-815, Enzo Life Sciences, Farmingdale, NY, USA). HRP-conjugated secondary antibodies were obtained from Jackson Immuno Research Labs (West Grove, PA, USA). The antibodies were diluted in a Can Get Signal solution (TOYOBO, Osaka, Japan).

### 2.10. Measurement of Oxidative Stress and Plasma Antioxidant Capacity

Plasma samples were collected from mice to measure oxidative stress and antioxidant capacity in the peripheral blood. Briefly, plasma samples were mixed with Diacron-reactive oxygen metabolites (d-ROM) or biological antioxidant potential (BAP) test reagents (WISMERLL Co., Tokyo, Japan), following the recommended protocol, and absorbance was measured. This quantified the degree of oxidative stress and antioxidant capacity of the plasma samples. The measurements of the d-ROM or BAP tests were represented by Carratelli units “CARR U” (1 CARR U = 0.08 mg H_2_O_2_/dL) or μmol/L, respectively.

### 2.11. Immunohistochemistry (IHC) and Quantification of Infiltration of Immune Cells

Murine lung tissue specimens were fixed in formaldehyde and then embedded in paraffin. After sectioning (3 μM thick), paraffin-embedded lung tissue sections were immersed in EDTA solution (pH 8.0) (for Iba1 and LY6G) or antigen retrieval solution pH 9.0 (Nichirei Bioscience, Tokyo, Japan) (for CD3) and heated in a pressure cooker for immunostaining with anti-Iba-1 (for macrophage: 019-19741, WAKO, Osaka, Japan), anti-Ly6G (for neutrophils: ab238132, Abcam, Cambridge, MA, USA), and anti-CD3 (for T cells: 413591, Nichirei) antibodies. The sections were subsequently treated with HRP-conjugated goat anti-rabbit antibody (Nichirei, Tokyo, Japan), and the reactions were visualized with diaminobenzidine substrate system (Nichirei, Tokyo, Japan). The percentage of Iba-1+, Ly6G+, and CD3+ cells within the lung tissue was quantified using HALO software (Indica Labs, Corrales, NM, USA). Scoring was performed in a blinded fashion by a pathologist.

### 2.12. Statistical Analysis

For quantitative analysis, the data were presented as the mean ± SEM, and the statistical differences were analyzed by either a student’s *t*-test or the Tukey–Kramer test using Graph Pad Prism9 (Graph Pad Software Inc., San Diego, CA, USA), as indicated in each figure legend. The level of statistical significance was set at *p* < 0.05. The effect size was calculated following Cohen’s formula [24]. In some comparisons, effect size (*d*) is also determined. *d* = 0.2 was considered as a small effect, *d* = 0.5 as a medium effect, and *d* = 0.8 as a large effect.

## 3. Results

### 3.1. Intratracheal Administration of T_3_ Improves Pulmonary Pathology in Elastase-Induced COPD Mouse Model

In order to assess the T_3_ requirement in the lungs of the elastase-induced COPD model mice, simulating emphysema-dominant COPD, we examined the mRNA expression levels of *Dio2* [18], an enzyme that converts T_4_ into active T_3_, in the lung tissues. We found that the expression level of *Dio2* significantly increased in the lungs of the elastase-induced COPD mice on day 1, but this was not the case on day 4 after elastase administration (Figure 1A, Appendix A). The change in *Dio2* gene expression is similar to that of the IPF model mice in a previous report [20], implying that T_3_ requirement is transiently increased in the lungs of elastase-induced COPD model mice. On the other hand, elastase-induced *Dio2* up-regulation did not reflect the changes in protein expression both on days 1 and 4 (Appendix A), with one showing that the intratracheal administration of T_3_ improved pulmonary pathology in the IPF model mice [20] and the other showing that T_3_ indirectly played a pulmonary protective role in the acute lung injury (ALI) model mice [25]; it is possible that the intratracheal administration of T_3_ to elastase-induced COPD mice improves COPD pathology. In order to test this possibility, we intratracheally administered T_3_ (40, 80 μg/kg, every other day) to elastase-induced COPD model mice for 3 weeks and assessed the presence of emphysema and measured the respiratory parameters (inspiratory capacity (IC), compliance, elastance, forced expiratory volume in 0.1 s (FEV0.1), forced vital capacity (FVC), and forced expiratory volume % in 0.1 s (FEV0.1/FVC)) (Figure 1B). Emphysema was evaluated by mean linear intercepts (MLI), which is a general procedure for assessing emphysema, and respiratory function was measured by using the flexiVent system. The administration of T_3_ (80 μg/kg) suppressed emphysema (Figure 1C,D) and improved IC and compliance (Figure 1E,F). Moreover, elastance and FEV0.1/FVC showed a slight improvement in T_3_ (80 μg/kg)-treated elastase-induced group (Elastance: *p* = 0.07, effect size d: 1.26; FEV0.1: *p* = 0.05, effect size d: 1.42; FEV0.1/FVC: *p* = 0.07, effect size d: 1.24) (Figure 1G–J). In addition, to objectively assess emphysema, we performed an auto-measurement of alveolar morphology using an automatic cell count system: BZ-Analyzer [11]. The administration of T_3_ (80 μg/kg) suppressed some alveolar parameters, such as alveolar area (control and vehicle-treated elastase-induced group *p* = 0.09, effect size d: 1.43), alveolar perimeter (*p* = 0.06, effect size d: 1.63) (major axis + minor axis)/2 (*p* = 0.05, effect size d: 1.69), and ferret diameter (Figure 1K–N). Although some of the comparisons did not show statistical significance (*p* > 0.05), they were determined to be large based on the Cohen criterion, which is based on effect size (*d* > 0.8 as a large effect) [24]. These results indicate that T_3_ has a lung protective effect in the elastase-induced COPD mouse model.

### 3.2. T_3_ Improves Pulmonary Pathology via the Ppargc1a-Gclm Pathway in the Lungs of the Elastase-Induced COPD Mouse Model

In order to elucidate the mechanism of COPD improvement via T_3_ in the elastase-induced COPD mouse model, we focused on a mitochondrial master regulator, *Ppargc1a*, which is also down-regulated in IPF model mice [20]. We first examined the expression level of *Ppargc1a* in the lungs of the mice treated with T_3_ (80 μg/kg) for 1 day (Figure 2A). We found that the mRNA expression level of *Ppargc1a* significantly decreased in the lungs of the elastase-induced COPD mice on day 1 after elastase administration (Figure 2B). Importantly, *Ppargc1a* mRNA increased in the lungs of T_3_-treated mice when compared to the vehicle-treated mice (Figure 2B). Although the expression of PGC1α was not affected by day 4 (after elastase administration), its expression was slightly increased (*p* = 0.18, effect size d: 1.15) in the elastase-induced COPD model treated with T_3_ when compared to the vehicle-treated mice (Figure 2C,D). Notably, the intratracheal administration of T_3_ (80 μg/kg) for 4 days slightly increased the mRNA expression of *Ppargc1a* when compared with vehicle (*p* = 0.20, effect size d: 1.62) (Appendix A). On the other hand, T_3_ treatment for 1 week or 3 weeks did not increase the mRNA expression of *Ppargc1a* when compared with vehicle treatment (Appendix A). Overall, these data suggest that T_3_ suppresses early injury induced by elastase treatment. Moreover, this result is in agreement with a previous report, wherein it was found that T_3_ inhibits acute lung injury in hyperoxia-induced acute lung injury model mice [26]. Based on a previous report that showed that T_3_ suppresses oxidative stress and inflammation [17], we next sought to examine oxidative and inflammatory markers. As illustrated in Figure 2E, we observed an increase in the mRNA level of the antioxidant factor *Gclm*. However, most oxidative stress-related factors (*Nrf2*, *Nqo1*, *Gsr*, *Gclc*) remained unaffected, with the exception of *Ho-1*, a target gene mainly induced by AP-1 and/or NF-κB [27] (Appendix A). The up-regulation of *Gclm*, which is a downstream factor of PGC1α, is consistent with the increase in PGC1α by T_3_ [28]. The intratracheal administration of T_3_ for 1 day or 4 days did not change the d-ROM level, which is an index of oxidative stress, and BAP, an index of antioxidant capacity, in the plasma (Figure 2F–I). This suggests that T_3_ acts only on the lungs. The expression level of *Kc*, an inflammatory cytokine, was not changed by the intratracheal administration of T_3_ (Figure 2J). In order to examine the infiltration of immune cells into lung tissues, we performed immuno-histochemical staining with anti-CD3 (T cell marker), IBA1 (macrophage marker), or Ly6G (neutrophil marker) antibody. The intratracheal administration of T_3_ (80 μg/kg) for 1 day did not affect the infiltration of neutrophils (Appendix A). This suggests that the effect of T_3_ is not an anti-inflammatory effect.

According to previous reports, thyroid hormone is an essential factor for lung development and growth [19], and some growth factors relate to the repair of lung tissues [29,30,31,32,33,34]. Therefore, we examined the expression level of those growth factors. T_3_ did not change the expression level of some growth factors in elastase-induced COPD model mice (Appendix A). These results indicate that the intratracheal administration of T_3_ ameliorates pathology via the *Ppargc1a*-*Gclm* pathway in the lungs of elastase-induced COPD mouse models in the early phase after elastase administration.

### 3.3. The Intratracheal Administration of T_3_ for 12 Days Did Not Improve COPD Pathology in C57BL/6J-βENaC-Tg Mice

We showed that the intratracheal administration of T_3_ improved pathology in the elastase-induced COPD model mice. However, whether T_3_ suppresses pathology in the C57BL/6J-βENaC-Tg mice that simulated airway-dominant COPD remains unknown. In order to assess the mitochondrial function and T_3_ requirement in the lungs of C57BL/6J-βENaC-Tg mice, we examined the expression levels of *Dio2* and *Ppargc1a* in the lung tissues with RT-qPCR. The expression level of *Dio2* mRNA slightly changed in the lungs of C57BL/6J-βENaC-Tg mice compared with the WT mice (*Dio2* WT and ENaC-Tg mice), *p* = 0.06, effect size d: 1.49) (Figure 3A), while *Ppargc1a* did not change (Figure 3B). In order to investigate the effect of T_3_ on C57BL/6J-βENaC-Tg mice, we administered T_3_ intratracheally (at doses of 40 or 80 μg/kg every other day) for a duration of 12 days. This was carried out on C57BL/6J-βENaC-Tg mice whose age and sex were matched with those of the elastase-treated mice (Figure 3C). We assessed the presence of emphysema and measured the respiratory function. In accordance with previous reports [12,13,14,15], the C57BL/6J-βENaC-Tg mice exhibited the emphysematous phenotype (Figure 3D,E) and alterations in the respiratory parameters, including FEV0.1 (Figure 3F, FEV0.1 (WT and ENaC-Tg mice), *p* = 0.11, effect size d: 2.23), FVC (Figure 3G), and FEV0.1/FVC (Figure 3H). However, the administration of T_3_ to C57BL/6J-βENaC-Tg mice did not improve the COPD phenotype (Figure 3D–H). The mRNA level of *Ppargc1a*, *Gclm*, and *Dio2*, as well as the protein level of PGC1α, in T_3_-treated C57BL/6J-βENaC-Tg mice did not change compared to the vehicle-treated C57BL/6J-βENaC-Tg mice (Figure 3I–K, Appendix A). These results indicate that T_3_ requirement (the expression level of *Dio2*) does not increase in C57BL/6J-βENaC-Tg mice and that T_3_ administration could not improve COPD pathology in these mice.

### 3.4. The Intratracheal Administration of T_3_ for 2 Months Did Not Improve COPD Pathology in C57BL/6J-βENaC-Tg Mice

The C57BL/6J-βENaC-Tg mice stably presented COPD pathology [11], so we intratracheally administered T_3_ into the C57BL/6J-βENaC-Tg mice for a longer time span. In order to reduce any possible damage to the trachea due to long-term intratracheal administration, we used aerosol sprayers. We intratracheally administered T_3_ (40 or 80 μg/kg every other day) for 2 months (Figure 4A), assessed the presence of emphysema, and measured the respiratory function. Consistently, the C57BL/6J-βENaC-Tg mice exhibited the emphysematous phenotype (MLI (WT and ENaC-Tg mice), *p* = 0.05, effect size d: 2.64) (Figure 4B,C) and alterations in the respiratory parameters, including FEV0.1 (Figure 4D, FEV0.1 (WT and ENaC-Tg mice), *p* = 0.31, effect size d: 1.67), FVC (Figure 4E, FVC (WT and ENaC-Tg mice), *p* = 0.09, effect size d: 2.28), and FEV0.1/FVC (Figure 4F, FEV0.1/FVC (WT and ENaC-Tg mice), *p* = 0.14, effect size d: 1.86). The administration of T_3_ into the C57BL/6J-βENaC-Tg mice did not improve the COPD phenotype (Figure 4B–F). Moreover, the mRNA level of *Ppargc1a*, *Gclm*, and *Dio2*, as well as the protein level of PGC1α in the T_3_-treated C57BL/6J-βENaC-Tg mice, did not change compared to the vehicle-treated C57BL/6J-βENaC-Tg mice (Figure 4G–I, Appendix A). These results indicate that the intratracheal administration of T_3_ for longer time spans does not improve COPD pathology in C57BL/6J-βENaC-Tg mice.

## 4. Discussion

Guoing Yu et al. have shown that T_3_ has the potential to be a novel therapeutic agent for IPF [20]. However, the role of T_3_ in COPD pathogenesis remains unclear and needs to be clarified. In this study, we found that the expression level of *Dio2* increased and *Ppargc1a* decreased in elastase-induced COPD mice (emphysema-dominant COPD model) but not in C57BL/6J-βENaC-Tg mice. *Dio2* and *Ppargc1a* are indexes of T_3_ requirement and mitochondrial function, respectively. Based on our data, we hypothesized that T_3_ was lacking in the lungs of elastase-induced COPD model mice, and we intratracheally administered T_3_. This improved pulmonary pathology in elastase-induced COPD model mice, likely via the *Ppargc1a*-*Gclm* pathway. The biochemical analysis suggested that this effect may be specifically at work during the first few days after inducing pathology. On the other hand, the intratracheal administration of T_3_ to C57BL/6J-βENaC-Tg mice (airway-dominant COPD model) did not improve COPD pathology possibly due to the molecular characters (endotype) that show lower T_3_ requirement and normal mitochondrial function. These results suggest the importance of developing COPD therapies based on phenotype (emphysema- or airway-dominant) and/or endotype classification when considering that the effect of T_3_ is different between the two COPD models.

The difference in the inhibitory effect of T_3_ between these COPD models may be due to the difference in the degree of inflammation. It has been reported that the expression of *Dio2* increases with inflammation [35], and in the elastase-induced COPD model mice, severe inflammation is rapidly induced [11]. Therefore, T_3_ requirement increased with severe inflammation in the early stages after inducing pathology, and the intratracheal administration of exogenous T_3_ may have suppressed pulmonary pathogenesis. On the other hand, the expression level of *Dio2* did not increase in C57BL/6J-βENaC-Tg mice due to chronic inflammation [12]. Therefore, it is unsurprising that the intratracheal administration of T_3_ did not suppress pulmonary pathogenesis. This suggests that the exogenous T_3_ supplementation could potentially serve as a therapeutic option for acute, but not chronic, COPD exacerbation. Given the lack of effective agents to control the symptoms of acute COPD exacerbation, a major cause of mortality in this disease, our findings hold significant clinical relevance.

We observed that the expression of *Dio2* in C57BL/6J-βENaC-Tg mice slightly decreased compared with WT mice (Figure 3B). *Dio2* is an enzyme that converts T_4_ into active T_3_ [18], and T_3_ controls systemic metabolism [16]. Therefore, another possible explanation for the lack of effect of T_3_ on C57BL/6J-βENaC-Tg mice is that the level of metabolism may be lower in the lungs of these mice, and the expression of *Dio2* may not necessarily implicate T_3_ requirement. Moreover, unlike in the elastase-induced COPD mice, the expression of *Dio2* did not change from T_3_ treatment for 12 days (Appendix A), and it slightly increased from T_3_ treatment for 2 months (vehicle- and T_3_ (80 μg/kg)-treated ENaC-Tg mice, *p* = 0.23, effect size d: 1.05) (Appendix A). Whether T_3_ treatment for longer than 2 months can improve metabolism in the lungs and COPD pathology in C57BL/6J-βENaC-Tg mice needs further investigation.

In this study, we intratracheally administered T_3_ (not intraperitoneally or orally) in order to deliver T_3_ into the lungs. T_3_ controls systemic metabolism, so if exogenous T_3_ is intraperitoneally or orally administered, there is a possibility of acute weight reduction or metabolism overactivation as a side effect of T_3_. It was previously reported that the intratracheal administration of T_3_ did not change the concentration of T_3_ in peripheral blood [20]; therefore, the side effect of T_3_ was negligible.

T_3_ has been reported to have various bioactivities, such as improving mitochondrial function [20], antioxidant effects [17], and anti-inflammatory effects [36]. In this study, and in agreement with the previous report [20], the expression level of *Ppargc1a* was increased by the intratracheal administration of T_3_, suggesting that the improvement in mitochondrial function is partly related to the suppression of pulmonary pathology. The increased *Gclm*, an antioxidant factor, via the intratracheal administration of T_3_ was similar to previous reports [17]. Moreover, PGC1α protein, which was increased via T_3_ treatment in elastase-induced COPD mice, is a transcriptional factor that regulates *Gclm* [28].

In this study, we found that T_3_ has a pulmonary protective effect in the elastase-induced model mice of COPD pathology. Considering that previous reports have shown that T_3_ suppresses the pathogenesis of IPF and acute lung injury [20,25], T_3_ may be applied as a novel therapy for emphysema-dominant COPD. Interestingly, it was reported that the expression of DIO2 was increased in the serum of COPD patients [22], and T_3_ was previously suggested as a drug candidate for pulmonary diseases [37]. Considering our results and those of previous studies, the expression of *Dio2* and *Ppargc1a* may be one of the biomarkers and endotype markers of COPD [6,38]. Although more detailed studies will be needed, T_3_ may also be useful as a novel therapy for diseases associated with severe inflammation.

## 5. Limitations

The primary limitation of our study is the lack of testing on other COPD models, including both emphysema-dominant and airway-dominant types. Notably, emphysema-dominant acute models have been reported. While the elastase-induced model is commonly used in COPD research, its reliance on the artificial exposure of enzymes to mice may not accurately replicate the human condition. In order to validate our findings, we could consider using other major models, such as the ozone-induced COPD model and the cigarette smoke-induced COPD model. Regarding airway-dominant models, especially those that mimic mucus obstruction, there has been a lack of development. In this context, the C57BL/6J-βENaC-Tg mice represent a unique model exhibiting these phenotypes. In any case, future studies should aim to compare the phenotype/endotype between mouse models and human samples. Additionally, our study does not address whether endogenous thyroid hormones contribute to protection against emphysema-dominant COPD. This aspect could be further clarified using *Dio2* knockout mice. Lastly, while our results suggest that the intratracheal administration of T3 suppresses COPD lung pathology via the *Ppargs1a*-*Gclm* pathway, the direct measurement of mitochondrial function or oxidative stress in lung tissues was not conducted. Evaluating mitochondrial function using an extracellular flux analyzer and measuring oxidative stress in lung tissues would further support this mechanism.

## 6. Conclusions

We found that T_3_ requirement and mitochondrial biogenesis activity were different between the elastase-induced COPD model, which demonstrated emphysema-dominant COPD, and the C57BL/6J-βENaC-Tg mice, which demonstrated airway-dominant COPD. In accordance with T_3_ requirement, the intratracheal administration of T_3_ improved COPD pathology via the *Ppargc1a*-*Gclm* pathway in the elastase-induced COPD mouse model. On the other hand, the intratracheal administration of T_3_ did not improve COPD pathology in the C57BL/6J-βENaC-Tg mice. This study highlights the relevance of considering not only COPD phenotypes but also COPD endotypes (expression levels of *Ppargc1a* and/or *Dio2*) in the research and development of better treatment approaches for COPD.

## Figures and Tables

**Figure 1 antioxidants-13-00030-f001:**
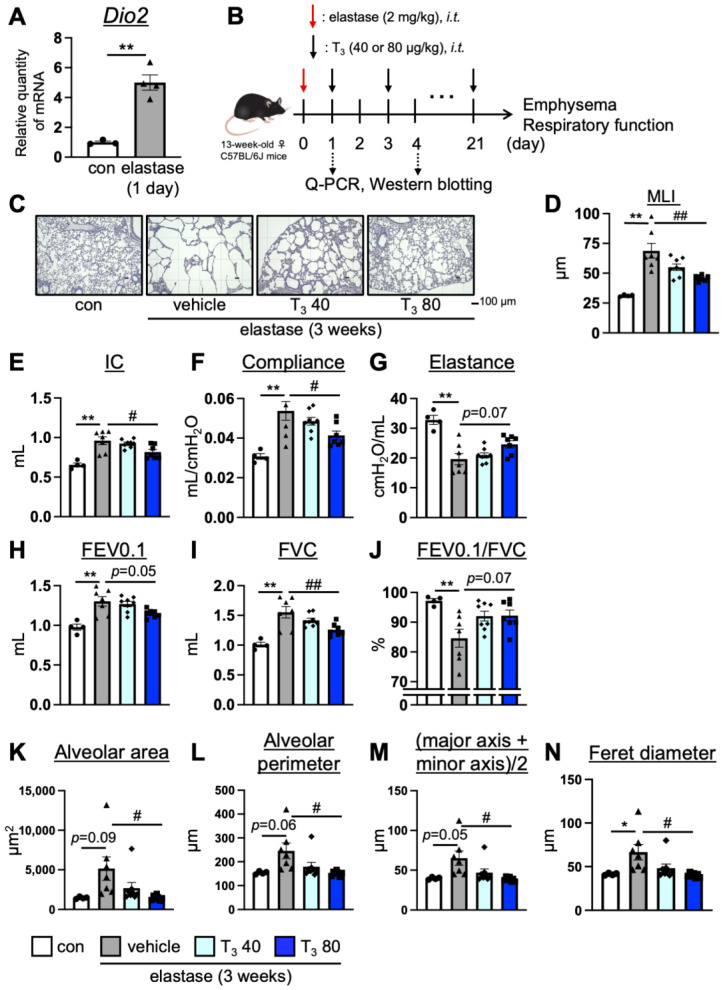
The effect of the intratracheal administration of T_3_ on the pulmonary pathology of COPD model mice. (**A**) The relative quantity of *Dio2* mRNA in the lungs of the control (white column and closed circle, *n* = 3) and elastase-induced COPD model mice (grey column and closed triangle, *n* = 4) (1 day after elastase administration) was measured with RT-qPCR. Data are means ± SEM ** *p* < 0.01 for Student’s *t*-test. (**B**) The experimental scheme of the intratracheal administration of T_3_ to elastase-induced COPD model mice. (**C**,**D**) Morphometric analysis of lungs in the control (white column and closed circle, *n* = 4), elastase-treated (grey column and closed triangle, *n* = 7), and intratracheally T_3_-administrated mice (40 μg/kg: light blue column and closed diamond; 80 μg/kg: blue column and closed square; *n* = 7–8, every other day for 3 weeks starting 1 day after elastase treatment) using MLI, as described in Methods. (**E**–**J**) Respiratory parameters of representative mice were analyzed with the flexiVent system. (**K**–**N**) Morphological alveolar parameters, such as alveolar area, alveolar perimeter, (major axis + minor axis)/2, and ferret diameter, were measured using an all-in-one fluorescence microscope—BZ-X710—and an automatic cell count system, as well as a BZ-analyzer in the representative mice. Data are means ± SEM * *p* < 0.05, ** *p* < 0.01 (vs. control), # *p* < 0.05, ## *p* < 0.01 (vs. elastase-vehicle): ANOVA with the Tukey–Kramer procedure.

**Figure 2 antioxidants-13-00030-f002:**
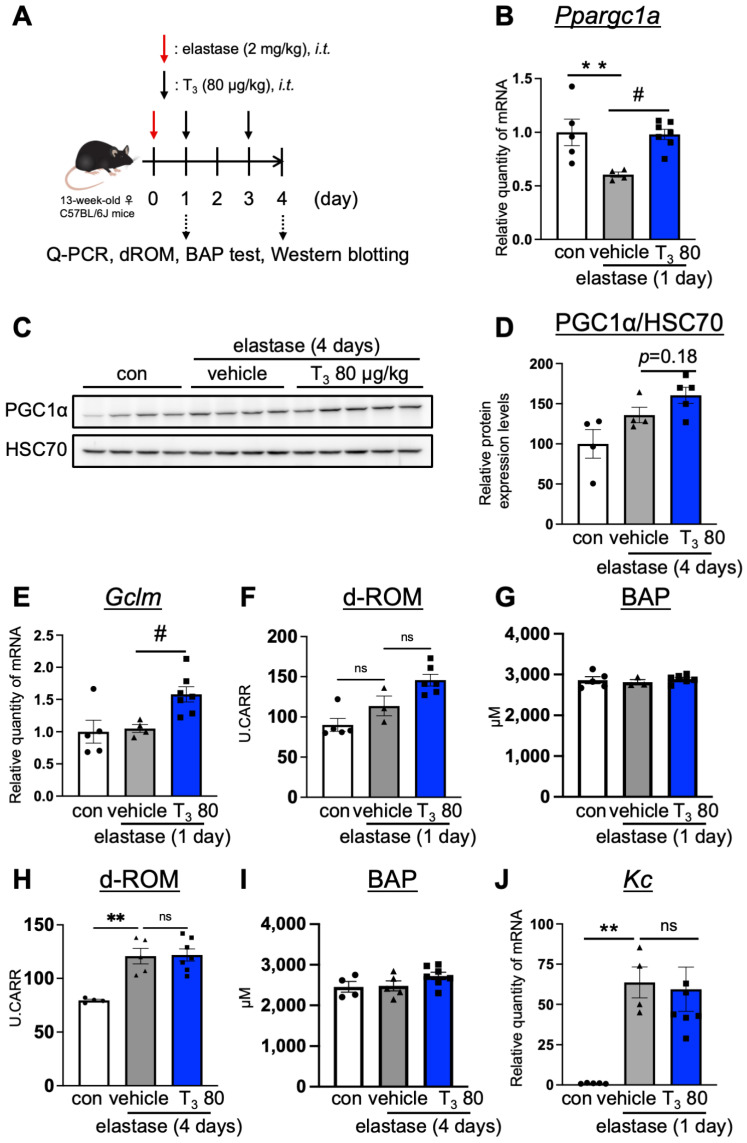
The mechanism of COPD improvement via T_3_ in elastase-induced COPD model mice. (**A**) The experimental scheme of the intratracheal administration of T_3_ into elastase-induced COPD model mice and the analysis performed. (**B**,**E**,**J**) The relative quantity of mRNA levels of *Ppargc1a*, *Gclm*, and *Kc* was measured in the lungs of the control, elastase-treated, and intratracheally T_3_-administered mice (80 μg/kg, one injection after 1 day of elastase treatment). (**C**,**D**) The protein expression levels of PGC1α in the lungs of the control, elastase-treated, and intratracheally T_3_-administered mice (80 μg/kg, every other day for 4 days after 1 day of elastase treatment) were assessed by immunoblotting. HSC70 was used as the loading control. The band intensity was quantified by using Multi Gauge software (FUJIFILM, Tokyo, Japan). (**F**–**I**) Oxidative stress and the antioxidant capacity were quantified via d-ROM and BAP tests, respectively, in the plasma of the control, elastase-treated, and intratracheally T_3_-administered mice (80 μg/kg, one injection 1 day after elastase treatment (**F**,**G**) or every other day for 4 days starting 1 day after elastase treatment (**H**,**I**)). White column and closed circle represent the control group, grey column and closed triangle represent elastase-treated control group, and blue column and closed square represent T_3_-administered mice with elastase-treated group. Data are means ± SEM; *n* = 4–7 mice/group. *p* values were assessed using ANOVA with the Tukey–Kramer procedure. ** *p* < 0.01 (vs. control), # *p* < 0.05 (vs. elastase-vehicle), *ns*: not significant. The effect size, d, was defined by Cohen’s calculation formula [24].

**Figure 3 antioxidants-13-00030-f003:**
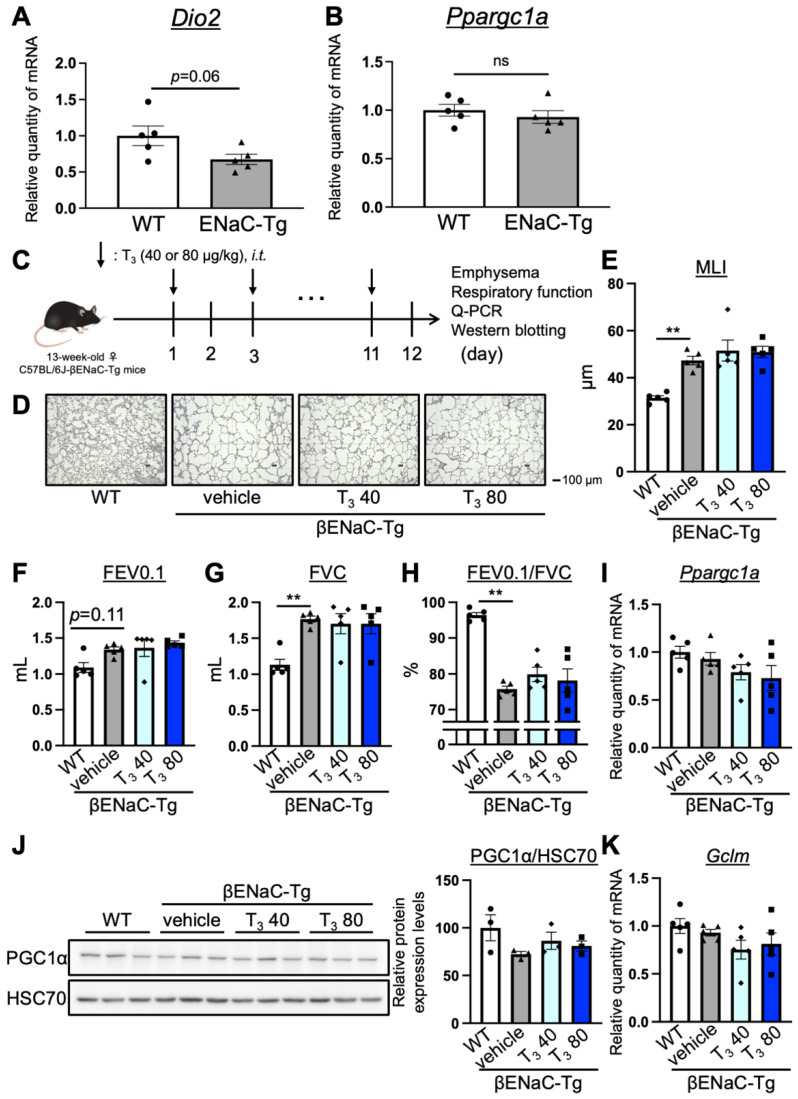
The effect of the intratracheal administration of T_3_ on the pulmonary pathology of C57BL/6J-βENaC-Tg mice. (**A**,**B**) The relative quantity of mRNA levels of *DIo2* and *Ppargc1a* was measured in the lungs of WT (white column and closed circle, *n* = 5) and C57BL/6J-βENaC-Tg mice (grey column and closed triangle, *n* = 5) with RT-qPCR. Data are means ± SEM for Student’s *t*-test. *ns*: not significant. (**C**) The experimental scheme of the intratracheal administration of T_3_ to C57BL/6J-βENaC-Tg mice. (**D**,**E**) Morphometric analysis of WT (white column and closed circle, *n* = 5), C57BL/6J-βENaC-Tg mice (grey column and closed triangle, *n* = 5), and intratracheally T_3_-administrated C57BL/6J-βENaC-Tg mice (40 μg/kg: light blue column and closed diamond; 80 μg/kg: blue column and closed square; *n* = 5, 40 or 80 μg/kg every other day for 12 days at 13-weeks-old) using MLI. (**F**–**H**) The respiratory parameters (FEV0.1, FVC, and FEV0.1/FVC) of the representative mice were analyzed using the flexiVent system. (**I**,**K**) The relative quantity of mRNA levels of *Ppargc1a* and *Gclm* was measured in the lungs of the representative mice using RT-qPCR. (**J**) The protein expression levels of PGC1α in the lungs of the representative mice were analyzed using immunoblotting (upper panel). HSC70 was used as the loading control. The band intensity was quantified by Multi Gauge software (FUJIFILM, Tokyo, Japan) (lower panel). Data are means ± SEM, ** *p* < 0.01 (vs. WT) using ANOVA with Tukey–Kramer procedure.

**Figure 4 antioxidants-13-00030-f004:**
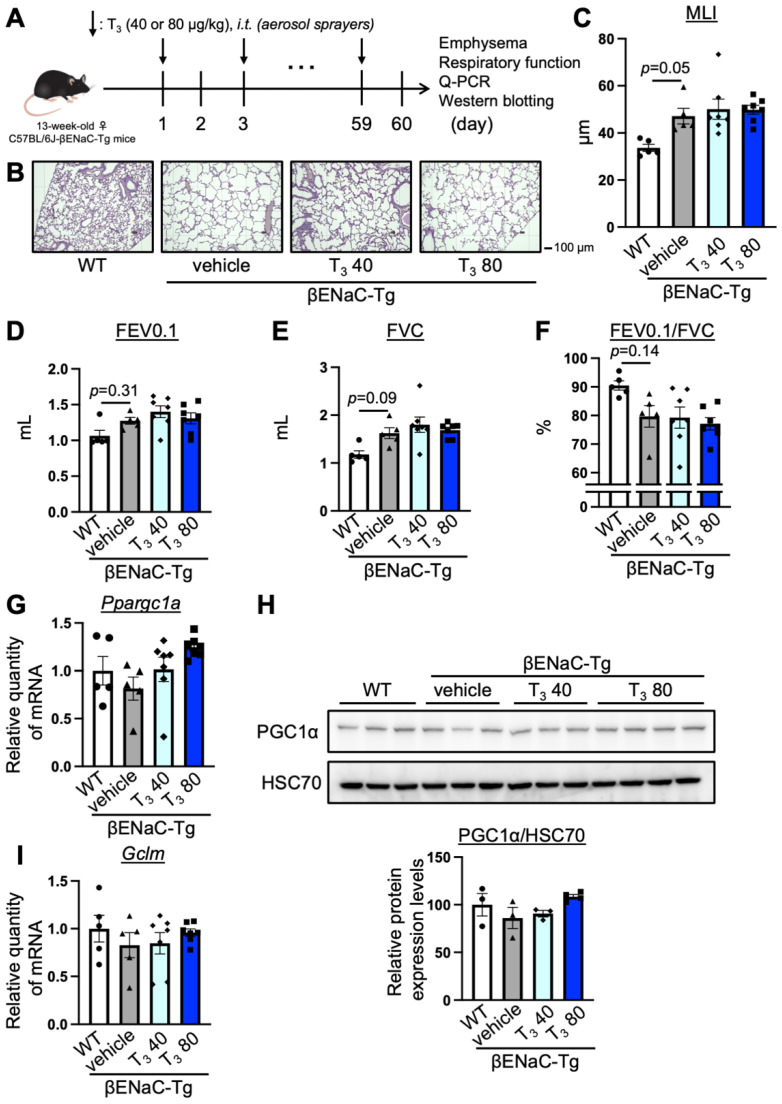
The effect of longer time span intratracheal administration of T_3_ on pulmonary pathology in C57BL/6J-βENaC-Tg mice. (**A**) The experimental scheme of the intratracheal administration of T_3_ into C57BL/6J-βENaC-Tg mice. (**B**,**C**) Morphometric analysis of WT (white column and closed circle, *n* = 5), βENaC-Tg mice (grey column and closed triangle, *n* = 5), and intratracheally T_3_-administrated C57BL/6J-βENaC-Tg mice (40 μg/kg: light blue column and closed diamond; 80 μg/kg: blue column and closed square; *n* = 5, 40 or 80 μg/kg every other day for 2 months at 13-weeks-old) using MLI. (**D**–**F**) Respiratory parameters (FEV0.1, FVC, FEV0.1/FVC) of representative mice were analyzed using the flexiVent system. (**G**,**I**) The relative quantity of mRNA levels of *Ppargc1a* and *Gclm* were measured in the lungs of the representative mice using RT-qPCR. (**H**) The protein expression levels of PGC1α in the lungs of the representative mice were analyzed by immunoblotting (upper panel). HSC70 was used as the loading control. The band intensity was quantified by Multi Gauge software (FUJIFILM, Tokyo, Japan) (lower panel). Data are means ± SEM from ANOVA using the Tukey–Kramer procedure.

**Table 1 antioxidants-13-00030-t001:** Primer sequences of genes for quantitative RT-PCR.

Primer	Gene ID	Sequence
*Dio2*_QRT-FW	13371	5′-TCAGGTAACAATTATGCCTCGGA-3′
*Dio2*_QRT-RV		5′-GCTGAACCAAAGTTGACCACC-3′
*Ppargc1a*_QRT-FW	19017	5′-AAGTGGTGTAGCGACCAATCG-3′
*Ppargc1a*_QRT-RV		5′-AATGAGGGCAATCCGTCTTCA-3′
*Il-8* (*Kc*)_QRT-FW	14825	5′-TGTCAGTGCCTGCAGACCAT-3′
*Il-8* (*Kc*)_QRT-RV		5′-CCTCGCGACCATTCTTGAGT-3′
*Gclm*_QRT-FW	14630	5′-CTTCGCCTCCGATTGAAGATG-3′
*Gclm*_QRT-RV		5′-AAAGGCAGTCAAATCTGGTGG-3′
*Nrf2*_QRT-FW	18024	5′-CACTCCAGCGAGCAGGCTAT-3′
*Nrf2*_QRT-RV		5′-CTGGGACTGTAGTCCTGGCG-3′
*Nqo1*_QRT-FW	18104	5′-TTCTGTGGCTTCCAGGTCTT-3′
*Nqo1*_QRT-RV		5′-AGGCTGCTTGGAGCAAAATA-3′
*Ho-1*_QRT-FW	15368	5′-GCCACCAAGGAGGTACACAT-3′
*Ho-1*_QRT-RV		5′-GCTTGTTGCGCTCTATCTCC-3′
*Gsr*_QRT-FW	14782	5′-CACGGCTATGCAACATTCGC-3′
*Gsr*_QRT-RV		5′-GTGTGGAGCGGTAAACTTTTTC-3′
*Gclc*_QRT-FW	14629	5′-GGACAAACCCCAACCATCC-3′
*Gclc*_QRT-RV		5′-GTTGAACTCAGACATCGTTCCT-3′
*Fgf2*_QRT-FW	14173	5′-GCGACCCACACGTCAAACTA-3′
*Fgf2*_QRT-RV		5′-TCCCTTGATAGACACAACTCCTC-3′
*Fgf7*_QRT-FW	14178	5′-ACCTGAGGATTGACAAACGAGG-3′
*Fgf7*_QRT-RV		5′-CCACGGTCCTGATTTCCATGA-3′
*Fgf10*_QRT-FW	14165	5′-GCAGGCAAATGTATGTGGCAT-3′
*Fgf10*_QRT-RV		5′-ATGTTTGGATCGTCATGGGGA-3′
*Igf1*_QRT-FW	16000	5′-CTACCAAAATGACCGCATCT-3′
*Igf1*_QRT-RV		5′-CAACACTCATCCACAATGCC-3′
*Tgfa*_QRT-FW	21802	5′-CACTCTGGGTACGTGGGTG-3′
*Tgfa*_QRT-RV		5′-CACAGGTGATAATGAGGACAGC-3′
*Hgf*_QRT-FW	15234	5′-AACAGGGGCTTTACGTTCACT-3′
*Hgf*_QRT-RV		5′-CGTCCCTTTATAGCTGCCTCC-3′
*Gapdh*_QRT-FW	14433	5′-CCTGGAGAAACCTGCCAAGTATG-3′
*Gapdh*_QRT-RV		5′-GGTCCTCAGTGTAGCCCAAGATG-3′
*18s*_QRT-FW	19791	5′-GTAACCCGTTGAACCCCATT-3′
*18s*_QRT-RV		5′-CCATCCAATCGGTAGTAGCG-3′

## Data Availability

Data is contained within the article.

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
