# Peer review of "T3 Intratracheal Therapy Alleviates Pulmonary Pathology in an Elastase-Induced Emphysema-Dominant COPD Mouse Model"

_antioxidants, 2023, doi:10.3390/antiox13010030_

Round 1

Reviewer 1 Report

Comments and Suggestions for Authors

The article titled "Thyroid Hormone T3 Improves COPD Pathology: A Tale of Two Models" provides valuable insights into the potential role of 3,3',5-Triiodo-L-Thyronine (T3) in the treatment of Chronic Obstructive Pulmonary Disease (COPD). The article presents a comprehensive study that addresses the differences in T3 requirements and the effects of mitochondrial biogenesis between two distinct COPD models, namely the emphysema-dominant and airway-dominant models. The positive aspects of this article are numerous, and it makes a significant contribution to the field of COPD research. However, there are some important points that should be addressed for a more comprehensive evaluation.

  1.  

Major Comments:

  1. - I found several english grammar errors throughout the manuscript. Please have a deep language revision.

  2.  
  3. - While the article focuses on animal models, it could benefit from a discussion of how the findings might translate to clinical settings. Are there potential implications for COPD treatment in humans? This connection to real-world applications would enhance the article's significance.

  4.  
  5. - The article should include more recent references to support the discussion. Since the most recent included reference is 2020, consider including studies or publications up to that date to provide a more up-to-date context for the research (if available).

  6.  
  7. -It would be beneficial to include a section on the limitations of the study. Highlighting potential sources of bias or areas where further research is needed would strengthen the article's scientific rigor.

  8.  
  9. -The article's significance to the broader field of COPD research should be emphasized. Discuss how these findings fit into the existing body of knowledge and suggest potential future directions for research in this area.

In conclusion, the article represents a positive contribution to the understanding of T3's potential role in improving COPD pathology. It provides a solid foundation for future research in this area. However, addressing the major comments mentioned above would further enhance the article's clarity and relevance

Comments on the Quality of English Language

moderate revision required 

Author Response

I wish to revise the Research Article for publication in Antioxidants, titled “T3 intratracheal therapy alleviates pulmonary pathology in an elastase-induced emphysema-dominant COPD mouse model.”

We are grateful for the comments and suggestions of the reviewers. The comments from the reviewers helped to vastly improve the paper.

Our responses to the reviewers’ question and the details of the changes we made according to their comments have been submitted.

Additionally, we would like to request permission to add two co-authors, Keisuke Kawano and Yukio Fujiwara, who have made substantial contributions to the additional experiments necessary for finalizing this revision. We hope for your understanding in this matter.

We are hopeful that our findings have some significant contribution to a large and diverse audience in the field of cellular and molecular biological sciences.

   Thank you for your time. We look forward to hearing from you.

Sincerely,

Tsuyoshi Shuto, Ph.D.

Answer to Reviewer 1's comments was described as follows. The answers to all the reviewers is also attached in case you need to check.

Reviewer 1

1) I found several English grammar errors throughout the manuscript. Please have a deep language revision.

We acknowledge that the initial draft of our manuscript, specifically in the abstract, introduction, results, and discussion sections, contained several grammatical errors and lacked logical coherence. We have thoroughly reviewed the entire text and made necessary revisions for clarity and accuracy. Additionally, we have introduced new figures and re-arranged existing ones for better representation. Parts of the text that have been re-written are highlighted in red. We believe these changes enhance the manuscript's precision and readability.

2) While the article focuses on animal models, it could benefit from a discussion of how the findings might translate to clinical settings. Are there potential implications for COPD treatment in humans? This connection to real-world applications would enhance the article's significance.

Thank you for your insightful comments regarding the clinical translational aspect of our study. We observed a specific up-regulation of the extrathyroidal T3-producing enzyme Dio2, considered as a marker of T3 requirement, in elastase-induced COPD lungs. These findings suggest that exogenous T3 supplementation could potentially serve as a therapeutic option for acute, but not chronic, COPD exacerbation. Given the lack of effective agents to control symptoms of acute COPD exacerbation, a major cause of mortality in this disease, our findings hold significant clinical relevance. Additionally, our study underscores the importance of considering not only COPD phenotypes but also COPD endotypes, as indicated by the expression levels of Ppargc1a and/or Dio2, in researching and developing more effective treatment approaches for COPD. We have emphasized these points in the revised manuscript, adding relevant sentences to the abstract (lines 26-41, marked in red), the discussion (lines 474-478, marked in red), and the conclusions (lines 539-541, marked in red).

3) The article should include more recent references to support the discussion. Since the most recent included reference is 2020, consider including studies or publications up to that date to provide a more up-to-date context for the research (if available).

We appreciate your suggestion regarding the inclusion of more recent references. In response to your feedback, we have revised the manuscript to emphasize the endotype theory, which has been a topic of recent discussions in the field. To support this, we have added references to up-to-date papers (see updated reference list, ref. 6 and 38).

4) It would be beneficial to include a section on the limitations of the study. Highlighting potential sources of bias or areas where further research is needed would strengthen the article's scientific rigor.

We appreciate your suggestion regarding the inclusion of a section on the study's limitations. The primary limitation of our study is the lack of testing on other COPD models, including both emphysema-dominant and airway-dominant types. Notably, emphysema-dominant acute models have been reported. While the elastase-induced model is commonly used in COPD research, its reliance on the artificial exposure of enzymes to mice may not accurately replicate the human condition. To validate our findings, we could consider using other major models such as the ozone-induced COPD model and the cigarette smoke-induced COPD model. Regarding airway-dominant models, especially those that mimic mucus obstruction, there has been a lack of development. In this context, the C57BL/6J-βENaC-Tg mice are a unique model exhibiting these phenotypes. In any case, future studies should aim to compare the phenotype/endotype between mouse models and human samples. Additionally, our study does not address whether endogenous thyroid hormones contribute to protection against emphysema-dominant COPD. This aspect could be further clarified using Dio2 knockout mice. Lastly, while our results suggest that intratracheal administration of T3 suppresses COPD lung pathology via the Ppargs1a-Gclm pathway, direct measurement of mitochondrial function or oxidative stress in lung tissues was not conducted. Evaluating mitochondrial function using an extracellular flux analyzer and measuring oxidative stress in lung tissues would further support this mechanism. These points have been incorporated into the revised manuscript in the limitations section (lines 514-531, marked in red)."

5) The article's significance to the broader field of COPD research should be emphasized. Discuss how these findings fit into the existing body of knowledge and suggest potential future directions for research in this area.

The concept of 'Treatable Traits' has recently been proposed as a new paradigm in the management of airway diseases, particularly for complex diseases, aiming to apply personalized medicine to improve individual outcomes. Transitioning new treatment approaches from theoretical concepts to practical applications is challenging but necessary. As previously mentioned in our response to question 2, our study demonstrates that exogenous T3 supplementation may have therapeutic potential for acute, but not chronic, COPD exacerbation. Additionally, our findings underscore the importance of considering not just COPD phenotypes but also COPD endotypes, marked by the expression levels of Ppargc1a and/or Dio2, in the research and development of more effective treatment approaches for COPD. These insights significantly advance the relatively new concept of 'Treatable Traits' and should be key considerations in future COPD research. We have incorporated these points into the revised manuscript, specifically in the abstract (lines 26-41, marked in red), the discussion (lines 474-478, marked in red), and the conclusions (lines 539-541, marked in red).

Reviewer 2 Report

Comments and Suggestions for Authors

The current manuscript from Dr. Shuto’s research group explored the role of thyroid hormone (T3) in the pathogenesis of chronic obstructive pulmonary disease (COPD) using elastase-induced COPD model and C57BL/6-βENaC-Tg mice as the mouse models corresponding to emphysema- or airway-dominant COPD mouse models.

Authors observed improvements of emphysematous symptoms in the former model by T3 treatment but did not notice any effect of the treatment in the second model.

The observations are interesting, and in agreement with other related studies performed on similar topics. Authors are requested to address the following suggestions.

Suggestions

1.     Authors are reporting that Ppargc1a and Dio2 gene expression are respectively decreasing and increasing after 1 day following elastase administration (Figure 1A, B). Please report whether any change in protein expression was noted at this time point. Authors need this information to support the statement “These results suggest that mitochondrial function …elastase-induced COPD model mice”.

2.     Figure 2C, please explain what the “d” values mean. Authors stated that “PGC1α protein was significantly increased in the elastase-induced COPD model mice”, but the p-values are not reported in the graph.

3.     It seems that authors are using Hsp70 protein expression as loading control for immunoblots, please provide a justification for this.

4.     Supplementary Figure 2C, it is interesting to see that HO-1 expression was increased by elastase treatment. Please expand discussion on this.

5.     Authors compared C57BL/6J-βENaC-Tg mice with wild type C57BL/6J mice for mitochondrial function and T3 requirement. However, it is important that sex and age of the animals from these two groups match. Please confirm that the animals were matched properly. This is critically important, and authors should attempt to confirm that proper methodology was followed for these comparisons. A suggestion would be to follow the emphysematous lung lesions in a time-dependent manner in the Tg mice only, before T3 treatment.

6.     If possible, please consider including another morphometric marker of emphysema assessment, other than mean linear intercept.

Other comments

1.     Materials and Methods, 2.1. Animals, “mice were bred in light-dark cycle”, please explain what this statement means.

2.     When discussing the results of the study, please mention whether its gene expression or protein level that is referred.

3.     Please provide protein size markers next to the blots.

4.     Please define the abbreviations used in the manuscript at their first mention. For example, d-ROM and BAP.

5.     Authors are requested to include a “limitations” section in the discussion, as per their own discretion.

6.     The manuscript requires editing for language use.

Comments on the Quality of English Language

Moderate editing will be very helpful.

Author Response

I wish to revise the Research Article for publication in Antioxidants, titled “T3 intratracheal therapy alleviates pulmonary pathology in an elastase-induced emphysema-dominant COPD mouse model.”

We are grateful for the comments and suggestions of the reviewers. The comments from the reviewers helped to vastly improve the paper.

Our responses to the reviewers’ question and the details of the changes we made according to their comments have been submitted.

Additionally, we would like to request permission to add two co-authors, Keisuke Kawano and Yukio Fujiwara, who have made substantial contributions to the additional experiments necessary for finalizing this revision. We hope for your understanding in this matter.

We are hopeful that our findings have some significant contribution to a large and diverse audience in the field of cellular and molecular biological sciences.

Thank you for your time. We look forward to hearing from you.

Sincerely,

Tsuyoshi Shuto, Ph.D.

Answer to Reviewer 2's comments was described as follows. The answers to all the reviewers is also attached in case you need to check.

Reviewer 2

  • Authors are reporting that Ppargc1a and Dio2 gene expression are respectively decreasing and increasing after 1 day following elastase administration (Figure 1A, B). Please report whether any change in protein expression was noted at this time point. Authors need this information to support the statement “These results suggest that mitochondrial function …elastase-induced COPD model mice”.

Thank you for your comment. We observed a significant increase in Dio2 gene expression in the lungs of elastase-induced COPD mice on day 1, which was not maintained on day 4 post-elastase administration (Figure 1A, Supplementary Figure 1A, B). This pattern mirrors the transient increase in T3 requirement seen in the lungs of elastase-induced COPD model mice, similar to findings in IPF model mice from a previous report (ref. 20). However, this up-regulation of Dio2 at the mRNA level did not correspond to changes in protein expression at either day 1 or day 4 (Supplementary Figure 1C-F). The noted discrepancy between mRNA and protein levels of Dio2/DIO2 in our study could be due to the timing of sampling, as Dio2 up-regulation was transient. Additionally, previous studies have indicated an increase in DIO2 expression in the serum of COPD patients (ref. 22), suggesting that serum DIO2 analysis is necessary to demonstrate functional up-regulation.

Regarding Ppargc1a, another focus of our study, we found a significant decrease in its mRNA expression in the lungs of elastase-induced COPD mice at day 1 post-elastase administration (Figure 2B). Notably, Ppargc1a mRNA levels increased in T3-treated mice compared to vehicle-treated mice (Figure 2B). Although PGC1α protein expression was not significantly altered at day 4 post-elastase administration, we observed a slight increase (p=0.18, effect size d: 1.15) in T3-treated elastase-induced COPD model mice compared to vehicle-treated mice (Figure 2C-D). This suggests that while baseline expression of PGC1α protein is not affected in COPD lungs, T3 treatment can enhance PGC1α levels at both mRNA and protein levels, indicating that Ppargc1a may also serve as a marker for T3responsiveness. Based on your valuable feedback, we have conducted thorough analyses on Dio2/DIO2 and Ppargc1a/PGC1α at both mRNA and protein levels, which has provided insightful information. These findings have been incorporated into the revised manuscript, specifically in the results (lines 221-228, 268-283, marked in red).

  • Figure 2C, please explain what the “d” values mean. Authors stated that “PGC1α protein was significantly increased in the elastase-induced COPD model mice”, but the p-values are not reported in the graph.

Thank you for pointing this out. We have now included the p-values in the corresponding figures. Additionally, where comparisons did not show statistical significance (p>0.05), we used Cohen's d formula as a measure of the biological effect size (refer to ref. 24). According to Cohen's criteria, a d value greater than 0.8 indicates a large effect size. This clarification has been added to the corresponding sections of the results, and the new text is highlighted in red.

  • It seems that authors are using Hsp70 protein expression as loading control for immunoblots, please provide a justification for this.

'Hsc70' is a well-known housekeeping gene that is constitutively expressed in the cytoplasm of cells. In contrast, 'Hsp70', also known as 'Hsp72', is a popular heat shock-inducible factor with functions similar to Hsc70. However, its expression is not ubiquitous or constitutive. Therefore, we used 'Hsc70' as an internal control, a practice commonly observed in many studies.

  • Supplementary Figure 2C, it is interesting to see that HO-1 expression was increased by elastase treatment. Please expand discussion on this.

Based on previous reports indicating that T3 suppresses oxidative stress and inflammation (ref. 17), we examined various oxidative and inflammatory markers. As illustrated in Figure 2E, we observed an increase in the mRNA level of the antioxidant factor Gclm. However, most oxidative stress-related factors (Nrf2, Nqo1, Gsr, Gclc) remained unaffected, with the exception of Ho-1, as shown in Supplementary Figure 3. The upregulation of Gclm, a downstream factor of PGC1α, aligns with the increase in PGC1α induced by T3(ref. 28). Conversely, Ho-1, primarily induced by AP-1 and/or NF-κB (ref. 27), exhibited different behavior from other oxidative stress-related factors. Notably, Ho-1 is generally considered a protective molecule. Therefore, the enhanced expression of Ho-1 observed in response to elastase and T3 treatment suggests a protective effect in the lungs in this experimental context. These ideas have been incorporated into the revised manuscript, specifically in the results (lines 285-290, marked in red).

  • Authors compared C57BL/6J-βENaC-Tg mice with wild type C57BL/6J mice for mitochondrial function and T3 However, it is important that sex and age of the animals from these two groups match. Please confirm that the animals were matched properly. This is critically important, and authors should attempt to confirm that proper methodology was followed for these comparisons. A suggestion would be to follow the emphysematous lung lesions in a time-dependent manner in the Tg mice only, before T3 treatment.

In our study, we exclusively used female mice for all experiments and ensured that the age at the start of administration was consistent at 13 weeks old. This approach was adopted because C57BL/6J-βENaC-Tg mice exhibit stable COPD pathology at this age. We took great care to ensure that both age and sex were appropriately matched between the elastase-treated mice and C57BL/6J-βENaC-Tg mice. This methodology has been detailed in the revised manuscript, specifically in the results section (lines 332-335, marked in red).

  • If possible, please consider including another morphometric marker of emphysema assessment, other than mean linear intercept.

In addition to mean linear intercept (MLI) analysis for emphysema assessment, we utilized an automatic cell count system, BZ-Analyzer (ref. 11), to objectively evaluate alveolar morphology. Using this system, we found that T3 administration (80 μg/kg) had an impact on several alveolar parameters. These include alveolar area (control and vehicle-treated elastase-induced group p=0.09, effect size d: 1.43), alveolar perimeter (p=0.06, effect size d: 1.63), the average of the major and minor axis (p=0.05, effect size d: 1.69), and ferret diameter (Figure 1K-N). Although some comparisons did not reach statistical significance (p>0.05), the effect sizes were determined to be large based on Cohen's criteria (d>0.8 indicating a large effect) (ref. 24). We have incorporated these details into the revised manuscript, specifically in the materials and methods section (lines 158-162, marked in red) and the results section (lines 243-250, marked in red).

Other comments

1) Materials and Methods, 2.1. Animals, “mice were bred in light-dark cycle”, please explain what this statement means.

In our study, we utilized age-matched female C57BL/6J mice or C57BL/6J-βENaC-Tg mice for all animal experiments. These mice were maintained in a controlled environment with a 12-hour light/dark cycle and had ad libitum access to normal chow (provided by Oriental Yeast, Tokyo, Japan) in the animal facility at Kumamoto University. We have added this information to the revised manuscript, specifically in the materials and methods section (line 99, marked in red).

2) When discussing the results of the study, please mention whether its gene expression or protein level that is referred.

We conducted analyses on Dio2/DIO2 and Ppargc1a/PGC1α at both the mRNA and protein levels. In our manuscript, we have used small italicized characters to denote mRNA (gene), while capital letters are used for proteins. We have carefully reviewed the entire text and made the necessary revisions to ensure this distinction is consistently applied throughout.

3) Please provide protein size markers next to the blots.

We have updated the Full-length gel images to include protein size markers next to the blots on the right.

4) Please define the abbreviations used in the manuscript at their first mention. For example, d-ROM and BAP.

We have carefully reviewed the manuscript and added definitions for all abbreviations at their first mention, including d-ROM (derivatives of reactive oxygen metabolites) and BAP (biological antioxidant potential), to ensure clarity and ease of understanding for the reader. These revisions have been made throughout the text.

5) Authors are requested to include a “limitations” section in the discussion, as per their own discretion.

As previously mentioned in response to Reviewer 1's query, we have included a 'limitations' section in the revised manuscript. This new section can be found on lines 514-531, and it is highlighted in red for easy identification.

6) The manuscript requires editing for language use.

In line with our response to Reviewer 1's comment, we have conducted a thorough review of the entire manuscript and made necessary revisions to improve clarity and accuracy. Furthermore, we have added new figures and re-arranged existing ones to enhance their representation. Sections of the text that have been rewritten are highlighted in red. We are confident that these modifications have significantly improved the precision and readability of the manuscript.

Reviewer 3 Report

Comments and Suggestions for Authors

The authors report that intratracheal T3 administration improves pulmonary pathology in an elastase-induced emphysema-dominant COPD mouse model, while T3 had no effect on the emphysema in C57BL/6-βENaC-Tg mice.  Further, some mechanistic insights such as upregulation of known regulators of mitochondrial Ppargc1a and oxidative stress-related Gclm afterT3 injection is confirmed.  In C57BL/6-βENaC-Tg mice, T3 treatment did not improve COPD pathology.

There are several issues which need to be considered:

1.      Abstract is not very clear and lay be shortened.

2.      The C57BL/6-βENaC-Tg mice is not and needs to be described, only see reference 11.

3.      The elastase induce emphysema should include day 1 analyses on epithelial damage with protein leak, cytokine release of DNA or other DAMPs with activation of DNA sensing pathways. The histology in Figure 1 does not allow to identify recruited inflammatory cells. Further, data from 4 mice per group without any repeat study does not provide robust data.

4.      The choice of the emphysema models should be motivated. The elastase model is acute injury model; less aggressive models should be considered such as CS or ozone exposure. Please discuss this issue as well the endotypes, I do not understand.

Comments on the Quality of English Language

Needs to be improved.

Author Response

I wish to revise the Research Article for publication in Antioxidants, titled “T3 intratracheal therapy alleviates pulmonary pathology in an elastase-induced emphysema-dominant COPD mouse model.”

We are grateful for the comments and suggestions of the reviewers. The comments from the reviewers helped to vastly improve the paper.

Our responses to the reviewers’ question and the details of the changes we made according to their comments have been submitted.

Additionally, we would like to request permission to add two co-authors, Keisuke Kawano and Yukio Fujiwara, who have made substantial contributions to the additional experiments necessary for finalizing this revision. We hope for your understanding in this matter.

We are hopeful that our findings have some significant contribution to a large and diverse audience in the field of cellular and molecular biological sciences.

Thank you for your time. We look forward to hearing from you.

Sincerely,

Tsuyoshi Shuto, Ph.D.

Answer to Reviewer 3's comments was described as follows. The answers to all the reviewers is also attached in case you need to check.

Reviewer 3

  • Abstract is not very clear and lay be shortened.

In response to Reviewer 1's feedback, we have thoroughly reviewed and revised the entire manuscript to enhance its clarity and accuracy. Specifically for the abstract, we have composed entirely new text to ensure that it is more concise and clearer, thereby improving the overall precision and readability of the manuscript.

  • The C57BL/6-βENaC-Tg mice is not and needs to be described, only see reference 11.

We, along with other researchers, have established a new COPD mouse model known as the C57BL/6J-βENaC-Tg mice (refer to ref. 12). This model overexpresses the β subunit of the epithelial sodium ion channel (ENaC), a key regulator of salt and water reabsorption, specifically in airway epithelial cells. The increased influx of sodium ions into the intracellular compartment reduces periciliary liquid and leads to mucus stasis. Additionally, this mouse model displays several chronic clinical symptoms of COPD, including airway inflammation, emphysema, and a decline in respiratory function. Notably, by utilizing C57BL/6J-βENaC-Tg mice and elastase-induced mice, which differ in phenotypes and endotypes, we have identified unique modulatory factors that positively and negatively influence COPD pathogenesis in a phenotype/endotype-specific manner (refs. 12-15). We have added this information to the revised manuscript, specifically in the introduction (lines 68-77, marked in red).

  • The elastase induce emphysema should include day 1 analyses on epithelial damage with protein leak, cytokine release of DNA or other DAMPs with activation of DNA sensing pathways. The histology in Figure 1 does not allow to identify recruited inflammatory cells. Further, data from 4 mice per group without any repeat study does not provide robust data.

Regarding the involvement of DNA sensing pathways, we assessed the expression of Cgas, Sting, and IFNb mRNA using RT-qPCR. However, these genes exhibited low expression levels in the lungs of our COPD models, rendering them mostly undetectable and unaltered by T3 treatment (see attached file).

To further examine immune cell infiltration in lung tissues, we conducted a quantitative analysis using immunohistochemical staining with antibodies specific to CD3 (a T cell marker), IBA1 (a macrophage marker), and Ly6G (a neutrophil marker). We found that intratracheal administration of T3 (80 μg/kg) for 1 day did not significantly affect neutrophil infiltration, as shown in Supplementary Figure 4. Based on these results, we concluded that anti-inflammatory mechanisms do not appear to be a primary factor in the therapeutic effect of T3. This information has been added to the revised manuscript, specifically in the results section (lines 296-300, marked in red).

Regarding the sample size, we agree that replicates are essential for robust in vivo experiments. In our study, most experiments were conducted with more than five (n=5). For certain RNA and protein experiments, due to limitations in experimental capacity, we had to work with a smaller number of replicates, such as n=3 or n=4. However, these smaller datasets displayed low variation and were sufficient to show significant alterations when present. Crucially, our main data, including the expression of Dio2/DIO2 and Ppargc1a/PGC1α at both mRNA and protein levels, as well as phenotypic assessments of emphysema and respiratory dysfunction, were replicated at least twice. We hope this addresses your concerns.

  • The choice of the emphysema models should be motivated. The elastase model is acute injury model; less aggressive models should be considered such as CS or ozone exposure. Please discuss this issue as well the endotypes, I do not understand.

Thank you for your suggestion, which has also been highlighted by other reviewers. In this study, our models were limited to C57BL/6J-βENaC-Tg mice and elastase-induced mice. We acknowledge that different models could present varying phenotypes and endotypes. However, due to facility limitations, we were unable to test other models, such as those involving chronic smoke or ozone exposure. To address this, we have added a 'limitations' section to our manuscript, specifically discussing these aspects. This section can be found on lines 514-531 and is highlighted in red for easy reference.

Round 2

Reviewer 1 Report

Comments and Suggestions for Authors

Authors replied to my  comments in a satisfatorily way. IMHO this manuscript can now be accepted. Moreover, I agree adding those two new authors who contributed to carry out the revised version.

Reviewer 2 Report

Comments and Suggestions for Authors

Authors addressed the comments.

Comments on the Quality of English Language

Minor editing should be enough.

Reviewer 3 Report

Comments and Suggestions for Authors

The  revised manuscript responded to most questions and improved in quality. in view of the severity of PPE induced emphysema the authors might comment on the body weight loss and mortality of the mice, and whether T3 administration had any effect on BW and survival.

Comments on the Quality of English Language

The style and typos have to be checked.